

# The tadpole of *Chiasmocleis altomontana* (Anura: Microhylidae)

Leandro B. C. Menezes[1], Marcos R. Severgnini[1], Tiago L. Pezzuti[2,3], Michel V. Garey[4] and Diogo B. Provete[5,6,7]

[1] Institute of Biosciences, Federal University of Mato Grosso do Sul (UFMS), Campo Grande, Mato Grosso do Sul, Brazil

[2] Departamento de Ciências Biológicas, Universidade Estadual Paulista, São José do Rio Preto, São Paulo, Brazil

[3] Instituto de Ciências Biológicas, Universidade Federal de Minas Gerais, Belo Horizonte, Minas Gerais, Brazil

[4] Instituto Latino-Americano de Ciências da Vida e da Natureza, Universidade Federal da da Integração Latino-Americana, Foz do Iguaçu, Paraná, Brazil

[5] Biodiversity Synthesis, German Centre for Integrative Biodiversity Research, Leipzig, Sachsen, Germany

[6] Gothenburg Global Biodiversity Centre, University of Gothenburg, Göteborg, Västra Götaland, Sweden

[7] Department of Ecology, Universidade Federal de Mato Grosso do Sul, Campo Grande, Mato Grosso do Sul, Brazil

Corresponding author
Diogo B. Provete,
diogo.provete@ufms.br

## ABSTRACT

**Background**. Describing the morphology of anuran larvae contributes towards filling gaps in taxonomy and natural history. This is especially relevant for explosive breeders, in which adults remain at the reproduction site for only a short period, while tadpoles may be more conspicuous. Here, we describe the larval external morphology and internal oral anatomy of the microhylid frog *Chiasmocleis altomontana* from near its type locality in southeastern Brazil.

**Methods**. We took 13 linear morphometric measurements of 11 tadpoles between stages 35 and 39 from four ponds. To compare the larval external morphology of the genus, we also built a morphospace based on the log-shape ratio of linear measurements provided by the original descriptions. To impute missing data, we used a trait imputation method that considered the phylogenetic relationships and a Brownian Motion model of multivariate trait evolution. Finally, we provide novel quantitative and qualitative data on the tadpole of *Chiasmocleis anatipes* based on museum specimens.

**Results**. The tadpole of *C. altomontana* is the largest among the Atlantic Forest species and the second largest of the genus, after *C. anatipes*. Like all species of the genus, the tadpole of *C. altomontana* is exotrophic, suspension feeder, Orton type II, and occurs in lentic temporary environments. Overall, *C. altomontana* shows subtle differences in dorsal fin height, dorsal fin insertion, tail tip, and snout in lateral view from its congeners. The spiracle of *C. anatipes* is ventral, single, covering the vent tube, as in all other microhylids. The species has a larger tail and interorbital distance than its congeners. All species of the Atlantic Forest clade are clustered together in the morphospace, suggesting little disparity, while those of the Amazonian clade are more spread, suggesting higher morphological disparity. *Chiasmocleis altomontana* was close to its sister species, *Chiasmocleis mantiqueira*, while *C. anatipes* differs greatly in terms of shape from the remaining species of the genus. Our results can improve our understanding of the morphological diversity of microhylid tadpoles and reduce the diversity shortfall in anuran larval forms.

# INTRODUCTION

The genus *Chiasmocleis* currently comprises 37 species distributed throughout tropical South America (*Frost, 2024*). The genus can be divided into two major clades: Amazonian and Atlantic Forest, with the Atlantic Forest clade having originated from Amazonian dispersal (*De Sá et al, 2019*). While the species description rate has increased steadily, with 16 species being described in the last 20 years (*Frost, 2024*), only 13 (35%) of all species had their tadpoles described so far. At least 15 species of the genus occur in the Atlantic Forest (*Rossa-Feres et al., 2017*; *De Sá et al, 2019*), with only five (33%) having their tadpoles described (*Provete et al., 2012*). The morphology of tadpoles follows the same pattern for most microhylids, with an Orton type II, suspension feeding, lentic, and exotrophic larvae (*Altig & McDiarmid, 1999b*). However, there are also endotrophic tadpoles in the genus, such as *Chiasmocleis antenori* and *Chiasmocleis carvalhoi* (*Peloso et al., 2014*). While it has been hypothesized a few years ago that *Chiasmocleis magnova* could be a direct developer (*Moravec & Köhler, 2007*; *Peloso et al., 2014*), recent evidence (*Pérez-Peña, Tapia-Del-Aguila & Gagliardi-Urrutia, 2021*) showed that it has free-swimming larvae. Doubt has also been casted on the external morphology of *Chiasmocleis anatipes* (*Walker & Duellman, 1974*), whose description mentioned it had "paired spiracles ventrolaterally", a trait only shared by pipoids. All these controversies highlight the need to improve our knowledge about larval morphology of the genus if their evolution is to be understood.

Besides the external morphology, other larval characteristics can help not only resolving systematic relationships, but also shed light on ecological relationships of the species. For example, buccopharyngeal anatomy can provide not only useful information about phylogenetic affinities of a species, but also inform about habitat and dietary resource use (*Wassersug, 1980*) that can complement field observations. However, the only species of the genus that had its internal oral features described so far was *Chiasmocleis leucosticta* (*Langone et al., 2007*).

*Chiasmocleis altomontana* was described based on specimens from Estação Ecológica do Bananal, state of São Paulo, southeastern Brazil (*Forlani et al., 2017*). The species is endemic to the Atlantic Forest and known only from the Serra do Mar range (*Forlani et al., 2017*). The species was recovered as sister to *Chiasmocleis mantiqueira* in the most comprehensive phylogenetic hypothesis of the genus to date (*De Sá et al, 2019*), from which it diverged around 5 Mya. The adults of the two sister species are very similar, but *C. altomontana* can be distinguished from it by having short membrane among fingers, less abundant dermal spines than *C. mantiqueira*, and by having the first palatal grove pigmented. Besides the adult description (*Forlani et al., 2017*) and its abundance at the type locality (*Zaher, Aguiar & Pombal, 2005*), no further information is available about this species. Here, we provide information about the larval external morphology and internal oral features of *C. altomontana*, along with novel data on reproductive phenology and habitat use. We also

provide for the first time quantitative and qualitative data on *C. anatipes* based on museum specimens that help solve a 50-year controversy.

## MATERIALS & METHODS

We sampled adult and larval frogs and predaceous macroinvertebrates during a survey in 17 lentic water bodies using hand dipnets in the Serra da Bocaina National Park, Southeastern Brazil, from April 2008 to February 2010. We recorded adults, tadpoles, or spawns of *C. altomontana* in four temporary, closed-canopy ponds (Pond 1: −22.735133°S, −44.617133°W; Pond 2: −22.735078°S, −44.617219°W; Pond 3: −22.724109°S, −44.623817°W; Pond 4: −22.751018°S, −44.618552°W; Fig. 1). Unfortunately, we did not collect any tissue samples to conduct molecular identification of the tadpoles at the time of collection. Given that the lot of tadpoles has been stored in 10% formalin for more than 10 years, it was not possible to extract relevant DNA material for a molecular identification either. However, it is noteworthy that *C. altomontana* is the only species of this genus known to occur in the area in more than 60 years of surveys (*Zaher, Aguiar & Pombal, 2005*; *Serafim et al., 2008*; *Garey et al., 2014*). We encountered two calling males of *C. altomontana* in November 2008 in Pond 1. However, the advertisement call of these individuals was not recorded due to heavy rainfall. We found several bubble nests in the same pond the next day, which we attributed to *C. altomontana* given we encountered calling adults of this species in the site in the previous day. Subsequently, we encountered *Chiasmocleis* tadpoles in the same location, which we attributed to *C. altomontana* for the same reason. Both adults and tadpoles we found in this pond were collected and are now housed in the DZSJRP and CFBH collections (Appendix S1). ICMBio provided collecting permits (#14474-1, 14861-1, and 16461-1). At the time of sampling (2008 and 2009), the Brazilian law did not require any further authorization as institutional animal care and use committees were not established yet, given that the National Council for Animal Experimentation–CONCEA was created by law in October 2008 (Law # 11.794, of 8 October 2008). Thus, field work, sampling, and transportation followed the IBAMA ordinance (Portaria #16 of 4 March 1994).

Description of the larval external morphology was based on 11 specimens between stages 35–39 (*Gosner, 1960*). To describe the external morphology, we took the following linear measurements from pictures of preserved specimens to the nearest 0.01 mm using TPSDig2 (*Rohlf, 2015*), following *Altig & McDiarmid (1999a)*: total length (TL), body length (BL), tail length (TAL), tail muscle width (TMW), interorbital distance (IOD), maximum tail height (MTH), and tail muscle height (TMH); while we followed *Lavilla & Scrocchi (1986)* for body height (BH), body width (BW), eye-snout distance (ESD), and eye diameter (ED), and *Grosjean (2005)* for dorsal fin height (DFH) and ventral fin height (VFH). For the description of the internal oral features, we dissected three specimens at Gosner 37 following the procedures, terminology, and protocol of *Wassersug (1976)*. Pictures of the buccopharyngeal area were taken in a Leica stereomicroscope (model M205C). All pictures taken by us of tadpoles of *C. altomontana* are available at MorphoBank (http://dx.doi.org/10.7934/P5511).

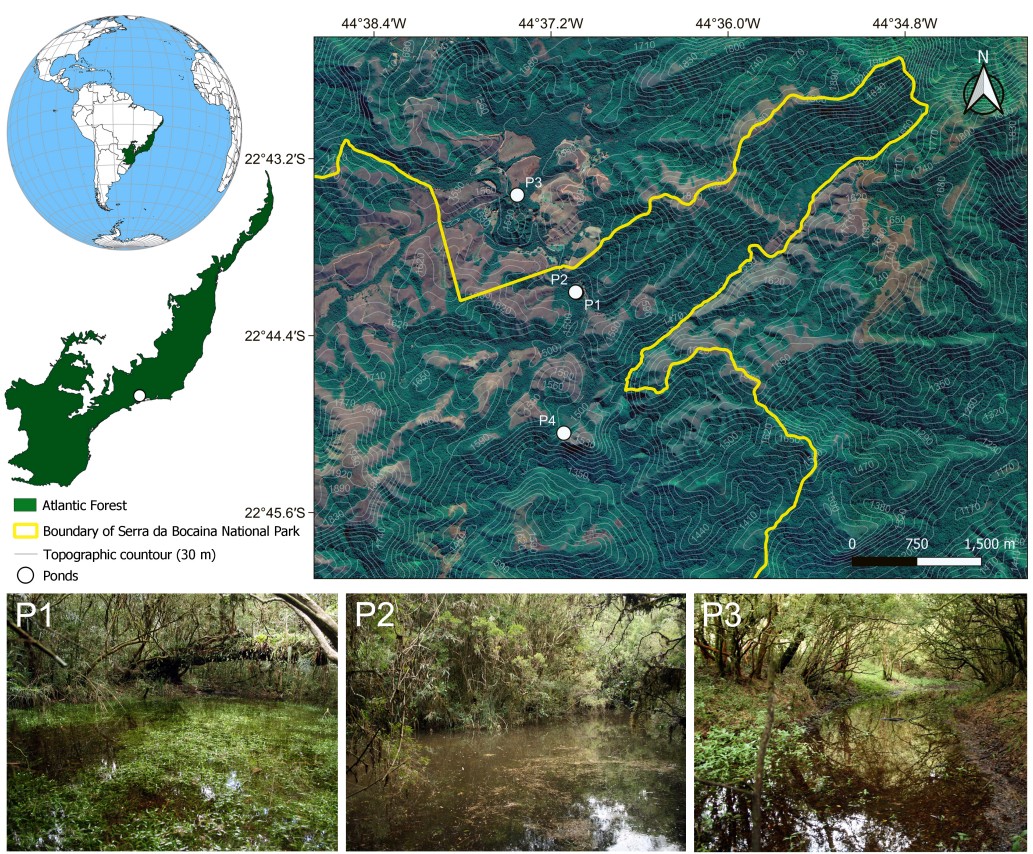

**Figure 1** **Map showing the four sampling sites in the Serra da Bocaina National Park, Southeastern Brazil.** Pictures of some of the ponds in which we collected tadpoles of *Chiasmocleis altomontana*. Contour lines indicate altitude quotas.

To compare the morphological diversity of tadpoles of the genus, we built a matrix with the same linear measurements as above for the 13 known tadpoles of the genus *Chiasmocleis* (*Menezes et al., 2024*). For species without published measurements, we took them from published line drawings using TPSDig2 (*Rohlf, 2015*) to the nearest 0.01 mm. We took measurements directly from five pictures of preserved specimens of *Chiasmocleis anatipes* (KU 146836) between stages 36 and 38 (available at MorphoBank, http://dx.doi.org/10.7934/P5511).

The percent of missing data in the whole matrix was 27.5%, with dorsal and ventral fin height, tail muscle width, and eye-snout distance missing 57% of the data. For further analysis, we removed these four variables with >50% of missing data, which brought the overall missing data down to 14.3%. The remaining nine variables had <30% of missing data, except for body height, which had 36%. This amount of data seems enough to produce reliable imputations (*Johnson et al., 2020*; *Jardim et al., 2021*). Then, we fitted alternative evolutionary models (single-rate Brownian Motion, single-peak Ornstein–Uhlenbeck, and Early Burst) to the data in the R package mvMORPH (*Clavel, Escarguel & Merceron, 2015*). The best-fitting model with the lowest corrected Akaike Information Criteria (AICc) and

Akaike weights = 1 was Brownian Motion (BM). Therefore, we applied an imputation method that used the phenotypic variance–covariance structure of the data under a BM model considering the species phylogenetic relationships to estimate missing data with the R package Rphylopars (*Goolsby, Bruggeman & Ané, 2016*). This method had good performance to impute missing data under several missing trait scenarios (*Penone et al., 2014*; *Johnson et al., 2020*), especially when traits are correlated in a phylogenetic context (*Jardim et al., 2021*), as in our case.

To remove the effect of size (*Klingenberg, 2016*) on linear measurements, we calculated the log-shape ratio (*Mosimann, 1970*; *Claude, 2013*) of the imputed linear measurements by calculating the geometric mean of each species, as a measure of size, and then dividing all variables by size. Finally, we conducted a Principal Components Analysis onto the matrix with log-shape ratios and took the first two eigenvectors to build the morphospace, superimposing the phylogeny (*De Sá et al, 2019*) on the reduced space in the R package geomorph (*Baken et al., 2021*; *Adams et al., 2024*). All analyses were performed in the R software v. 4.2.3 (*R Core Team, 2023*).

## RESULTS

### External morphology

The tadpole of *C. altomontana* is exotrophic, lentic and nektonic, with a total length of 19.5–24.57 mm (22.98 ± 1.45). The body length corresponds to 42.82% of the total length in Gosner stages 35–39 (Table 1). The body is depressed (BH/BW = 0.84–0.99), rounded in dorsal view (BL/BW = 1.40–1.45) and oval in lateral view (BL/BH = 1.47–1.67) at stage 36. The greatest body height is at the posterior third and the greatest width is at the middle of the body. Snout rounded in lateral and dorsal views (Figs. 2A, 2B). External nostrils are absent. Eyes laterally positioned. Nasolacrimal groove clearly visible in specimens from Gosner stages 32 to 38. Spiracle midventral, on the posterior part of the venter, opening broad, anterior to the vent tube, termination is free. Vent tube free, long, median, with medial aperture. Tail long (about 45–58% of total length), gradual thinning of the tail muscles, but not reaching the tail tip, which ends in a flagellum (Fig. 2A). Tail end is formed by a small fin membrane. The dorsal and ventral fins are individually larger than the musculature height in the beginning tail. Dorsal fin originating on the body, with an emergency angle of about 35°. Maximum height of the dorsal fin is at the middle third. Dorsal fin slightly lower than the ventral one. Margin of the dorsal and ventral fins in a wide arch (Table 2). Presence of a thickening of the dermis on the anterior portion of the tail.

Oral disc is terminal, with fleshy salience in the medial position of the lower jaw. Oral disc without papillae, jaw sheaths, or tooth rows. Oral disc with two symmetrical, semicircular labial flaps in front of the mouth, with acute tips, and separated by an inverted U-shaped medial notch.

### Coloration

In formalin, body coloration is formed by clusters of melanophores, denser on the dorsum and lateral of the body and tail musculature. These clusters are sparser on the venter and

**Table 1  Linear measurements of tadpoles of *Chiasmocleis altomontana* between stages 35 and 39 (*n* = 11).** Data are in millimeters and presented as mean ± standard deviation.

| Measurements | Stage 35 (*n* = 1) | Stage 36 (*n* = 2) | Stage 37 (*n* = 3) | Stage 38 (*n* = 3) | Stage 39 (*n* = 2) |
|---|---|---|---|---|---|
| TL | 8.9 | 22.55 ± 1.08 | 23.37 ± 0.75 | 23.35 ± 1.16 | 23.98 ± 0.83 |
| BL | 10.84 | 9.65 ± 0.09 | 9.95 ± 0.53 | 10.08 ± 0.25 | 9.99 ± 0.07 |
| TAL | 19.5 | 13.43 ± 1.29 | 13.61 ± 0.88 | 13.60 ± 1.27 | 14.27 ± 0.91 |
| MTH | 0.86 | 7.08 ± 0.02 | 7.29 ± 0.26 | 7.13 ± 0.20 | 6.36 ± 0.92 |
| TMH | 5.32 | 2.75 ± 0.18 | 2.77 ± 0.17 | 2.81 ± 0.29 | 2.83 ± 0.25 |
| TMW | 6.56 | 1.44 ± 0.09 | 1.80 ± 0.17 | 1.76 ± 0.06 | 1.73 ± 0.15 |
| IOD | 3.19 | 6.83 ± 0.14 | 6.88 ± 0.22 | 6.97 ± 0.06 | 6.81 ± 0.37 |
| BW | 2.44 | 6.78 ± 0.13 | 6.82 ± 0.29 | 6.78 ± 0.17 | 6.96 ± 0.17 |
| BH | 3.11 | 5.97 ± 0.11 | 6.18 ± 0.70 | 5.88 ± 0.35 | 5.83 ± 0.34 |
| ED | 6.07 | 0.84 ± 0.06 | 0.93 ± 0.05 | 0.91 ± 0.05 | 0.94 ± 0.04 |
| ESD | 6.34 | 4.41 ± 0.06 | 4.43 ± 0.16 | 4.53 ± 0.11 | 4.40 ± 0.41 |
| DFH | 1.74 | 2.44 ± 0.20 | 2.65 ± 0.44 | 2.73 ± 0.30 | 2.34 ± 0.30 |
| VFH | 4.21 | 2.72 ± 0.43 | 3.25 ± 0.10 | 2.91 ± 0.19 | 2.59 ± 0.07 |

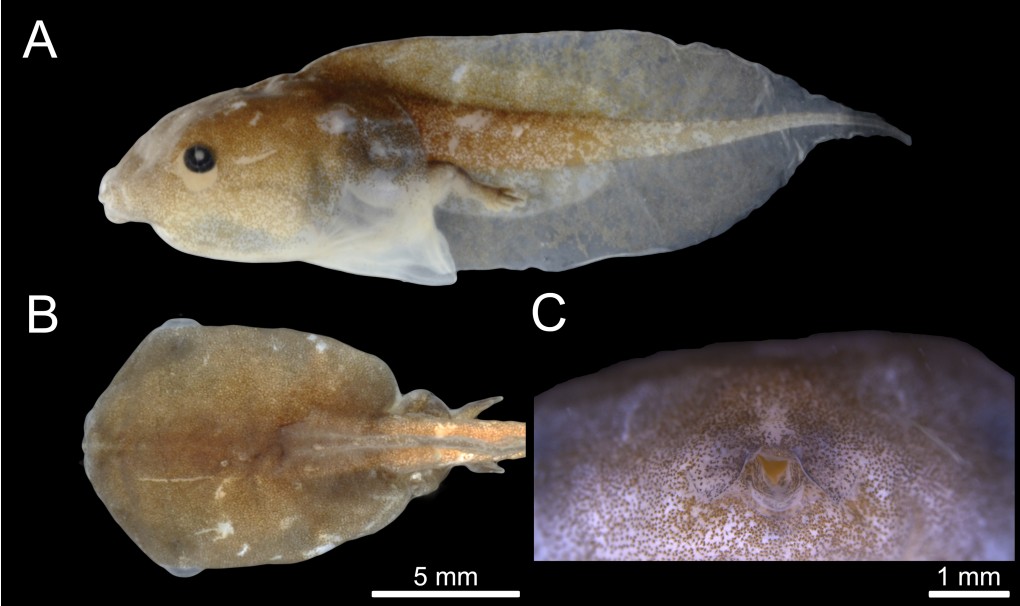

**Figure 2  External morphology of the tadpole of *Chiasmocleis altomontana* (DZSJRP- Tadpoles 2066.2) at Gosner Stage 38.** (A) Lateral view; (B) Dorsal view; (C) Frontal view of the oral apparatus (DZSJRP-Tadpoles 2054.1). Scale bar in (A) is the same as in (B).

fins, making the abdomen light brown and transparent on the fins, especially the flagellum, which is completely translucid. Iris black.

Menezes et al. (2025), *PeerJ*, DOI 10.7717/peerj.19220

**Table 2** **Comparison between morphological characters of *Chiasmocleis* larvae.** Terminology changed from original description to match that of *Pezzuti et al. (2021)*. Measurements are presented in millimeters.

| Species | Total length (stage) | Thickened basis of the tail | Dorsal fin insertion | Snout (lateral view) | Tail tip | Nasolacrimal grooves | References |
|---|---|---|---|---|---|---|---|
| *C. alagoana* | 17.40 (35–36) | Present | Body-tail junction | Truncated | Flagellum | Present | *Nascimento & Skuk, 2006* |
| *C. albopunctata* | 18.6 (38) | – | 1/3 of the body | Acuminated | Flagellum | – | *Oliveira-Filho & Giaretta, 2006* |
| *C. anatipes* | 31.5* | – | Body-tail junction | Rounded | Flagellum | – | *Walker & Duellman, 1974* |
| *C. antenori* | 7.14 (21) | – | Body-tail junction | Acuminated | Rounded | – | *Krügel & Richter, 1995* |
| *C. carvalhoi* | 17 (36) | – | – | Truncated | Flagellum | Present | *Wogel, Abrunhosa & Prado, 2004* |
| *C. hudsoni* | 11.47 (33) | – | Body-tail junction | Rounded | Acute | – | *Rodrigues et al., 2008* |
| *C. magnova* | 8.37 (29) | – | 1/3 of the body | Truncated | Rounded | – | *Pérez-Peña, Tapia-Del-Aguila & Gagliardi-Urrutia, 2021* |
| *C. lacrimae* | 18.49 (37) | – | 1/3 of the body | Truncated | Flagellum | Present | *Cordioli, Matias & Carvalho-e Silva, 2019* |
| *C. leucosticta* | 18.35 (34–37) | Present | At Body-tail junction | Inclined | Flagellum | Present | *Langone et al., 2007* |
| *C. mantiqueira* | 19.98 (36) | Present | At body-tail junction | Acuminated | Flagellum | Present | *Santana et al., 2012* |
| *C. schubarti* | 18.5 (34–38) | Present | – | Sloped | Flagellum | Not present | *Santos et al., 2015* |
| *C. shudikarensis* | 15.66 (36) | Present | At body-tail junction | Rounded | Flagellum | – | *Menin, Souza & Rodrigues, 2011* |
| *C. ventrimaculata* | 17 (36) | – | At body tail junction | Rounded | Flagellum | – | *Schlüter & Salas, 1991* |
| *C. altomontana* | 22.98 (35–39) | Present | Anteriorly to body-tail junction | Rounded | Flagellum | Present | This Study |

**Notes.**
*Description did not include Gosner stage of tadpoles.
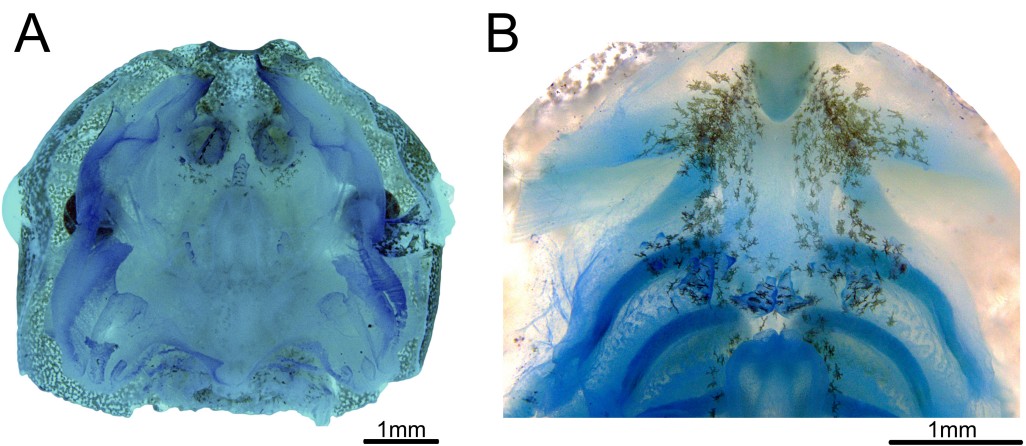

**Figure 3** **Buccopharyngeal morphology of the tadpole of *Chiasmocleis altomontana* at Gosner stage 37 (DZSJRP-Tadpoles 2054.1).** (A) Buccal roof; (B) buccal floor.

## Internal oral features

Buccal roof triangular shaped, as long as wide (Fig. 3A). Internal nares about 30% of the distance between the front of the mouth to esophagus. Median ridge about 70% of the distance between the front of the mouth to esophagus. The pernarial area has one or two pustulations medially. Four pairs of pustulations positioned laterally on the postnarial arena, varying in size from one side to the other. Nares length about 14% of the distance between the front of the mouth and esophagus. Distance between nares about half of their length. Nares rounded and oriented 15° from the transverse plane. Anterior wall not prominent. Prenarial papillae high, almost occluding nares; with serrated edges and conical shape, with broad bases. Four pairs of short, postnarial papillae projecting laterally above nares. Postnarial arena with many melanophores. Median ridge conical, longer than wide, with three pustulations on the anterior surface. Lateral ridge papillae absent. Buccal roof arena (BRA) oval. Two to four pairs of BRA papillae, which are short, with blunt tips, concentrated on the BRA's posterior end. No pustulations on the BRA. Dorsal velum discontinuous at midline. Glandular zone not distinct in the dorsal velum.

Buccal floor hexagonal, as long as wide (Fig. 3B). Lacking both infralabial and lingual papillae. Tongue anlage oval, positioned on a concave depression, lacking any papillae and pustulations. Buccal floor arena (BFA) trapezoidal. Two pairs of long, conical papillae lying on the posterior end of the BFA; papillae with serrated edges. About six pairs of pustulations close to BFA papillae, on the posterior end of the BFA. Four pairs of prepocket papillae. Buccal pockets oriented about 15°, about three times longer than wide. The glottis is fully visible from above (opening anteriorly), lying right posteriorly to the BFA papillae, and anteriorly to the ventral velum margins; lips tick. Esophageal funnel broad, slightly enlarged on the anterior end. The entire surface of the buccal floor is covered with scattered melanophores, more concentrated on the anterior end of the floor. Branchial baskets long. Ventral velum discontinuous at midline; lateral margins without peaks over cavities. Secretory pits and glandular zone overall indistinct.

## Natural history

*C. altomontana* has an explosive breeding strategy (*sensu Wells, 1977*), in which individuals reproduce only for a few days or weeks during the year. Adults of *C. altomontana* were recorded in breeding activity only in November 2008 in a temporary, closed-canopy pond. Males called on the leaf litter during the night at the pond margin. Adults of *Boana bandeirantes*, *Dendropsophus microps*, *D. minutus*, *Scinax* sp. (aff. *obtriangulatus*), and *Scinax hayii.* co-occurred with adults of *C. altomontana*. Spawns were deposited in bubble nests in lentic water bodies. The egg clutches contained an average of $223 \pm 42.4$ eggs ($n = 8$ clutches). Eggs had $1.44 \pm 0.5$ mm$^3$ ($n = 160$) of volume.

Tadpoles were found from January to April 2009, with peak abundance in February (313 individuals in Pond 2) and only one individual in April (Pond 2). Tadpoles of *C. altomontana* co-occurred with tadpoles of *Dendropsophus microps* and *Scinax hayii*. They also co-occurred with the following predatory invertebrates: adults of *Tropisternus* sp. (Coleoptera: Hydrophilidae), adults and larvae of *Cybister* sp. (Coleoptera: Dytiscidae), larvae of Dytiscidae, and larvae of *Rhionaeschna* (Odonata: Aeshnidae). Ponds in which the tadpole occurred had mean temperature 16.2 °C, pH 5.3, conductivity 0.11 mS/cm, turbidity 1 NTU, dissolved oxygen 2.19 (mg/L), depth between 0.34 and 0.8 cm, 94% of canopy cover, floating vegetation covering between 65 and 75% of the pond surface, and area between 29 and 88 m$^2$. The following items were found in the first centimeter of the intestines of tadpole: diatoms, *Phormidium* sp. (Cyanophyceae: Oscillatoriaceae), and *Trachelomonas* sp. (Euglenoidea: Euglenaceae).

## Morphospace

The majority of *Chiasmocleis* species of the Atlantic Forest clade were clustered together at the centre of the phylomorphospace (Fig. 4). The PC1 separated species with high tail muscle on the positive side, from those with low tail muscle on the negative side. PC2 separated species with longer interorbital distance and smaller tails on the positive side, from those with shorter interorbital distance and longer tails on the negative side. *C. altomontana* is on the quadrant with positive PC2 and negative PC1, close to *C. lacrimae, C. schubarti,* and its sister species, *C. mantiqueira*. Conversely, Amazonian species were widely distributed in the phylomorphospace, suggesting a higher morphological disparity than those from the Atlantic Forest. Interestingly, *C. hudsoni* is positioned within the species of the Atlantic Forest clade, far from its congeners of the Amazonian clade. The morphospace confirms the distinctive morphology of two species: *C. anatipes* (Fig. 5) appears far from the other species, at the extreme positive side of PC1, and the direct-developing species, *C. antenori*, at the extreme of both axes. Upon examination of preserved specimens of *C. anatipes*, we confirmed that its spiracle is single, ventral, median, with a large opening, covering the vent tube (Fig. 5). This is a character state shared by other species of the genus and different from the original description of the species (*Walker & Duellman, 1974*).

## DISCUSSION

Here, we described the 14th tadpole of the genus *Chiasmocleis* and the 6th from the Atlantic Forest clade (*Provete et al., 2012*). The tadpole of *C. altomontana* can be distinguished

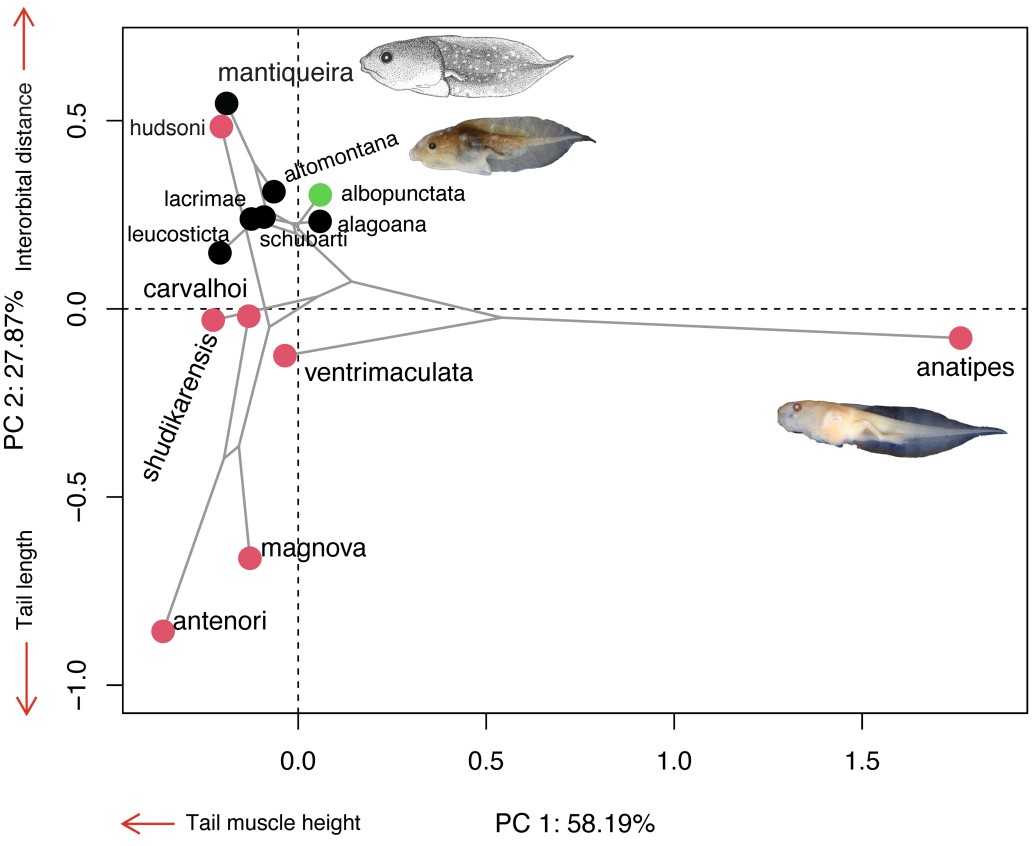

**Figure 4** **Ordination diagram showing the results of a principal component analysis (PCA) conducted on log-shape ratio variables with the distribution of 13 known tadpoles of the genus *Chiasmocleis*, with the phylogeny superimposed.** Colours represent biomes in which species occur: Atlantic Forest (black), Cerrado/Chaco (green), and Amazon (red). Illustrations taken from the original publications. Variables along the axis represent those with loadings >0.72.

from its congeners based on its dorsal fin insertion, total length, snout in lateral view, tail tip, and presence of nasolacrimal grooves (Table 2). *C. altomontana* has the second largest tadpole of the genus and the largest within the Atlantic Forest clade. The body of *C. altomontana* is approximately 42% of total length, while *C. antenori* has the smallest (29%) and *C. albopunctata* the largest (47%) body-tail proportion. Dorsal fin insertion is at the body-tail junction in *C. mantiqueira* (*Santana et al., 2012*), but anteriorly to the body-tail junction in *C. altomontana*). Although snout shape is somewhat an arbitrary character, *C. altomontana* has a rounded snout in lateral view, which is similar to *C. anatipes* (*Walker & Duellman, 1974*, Fig. 5), *C. hudsoni*, *C. shudikarensis*, and *C. ventrimaculata* (*Schlüter & Salas, 1991*; *Rodrigues et al., 2008*; *Menin, Souza & Rodrigues, 2011*). Nasolacrimal grooves are present in all stages analyzed of *C. altomontana*, also present at some stage in *C. alagoana* (*Nascimento & Skuk, 2006*), *C. leucosticta* (*Langone et al., 2007*), and *C. mantiqueira* (*Santana et al., 2012*; *Pezzuti et al., 2021*), all belonging the Atlantic Forest clade. However, most previous studies did not mention nasolacrimal grooves and more species may display this character.

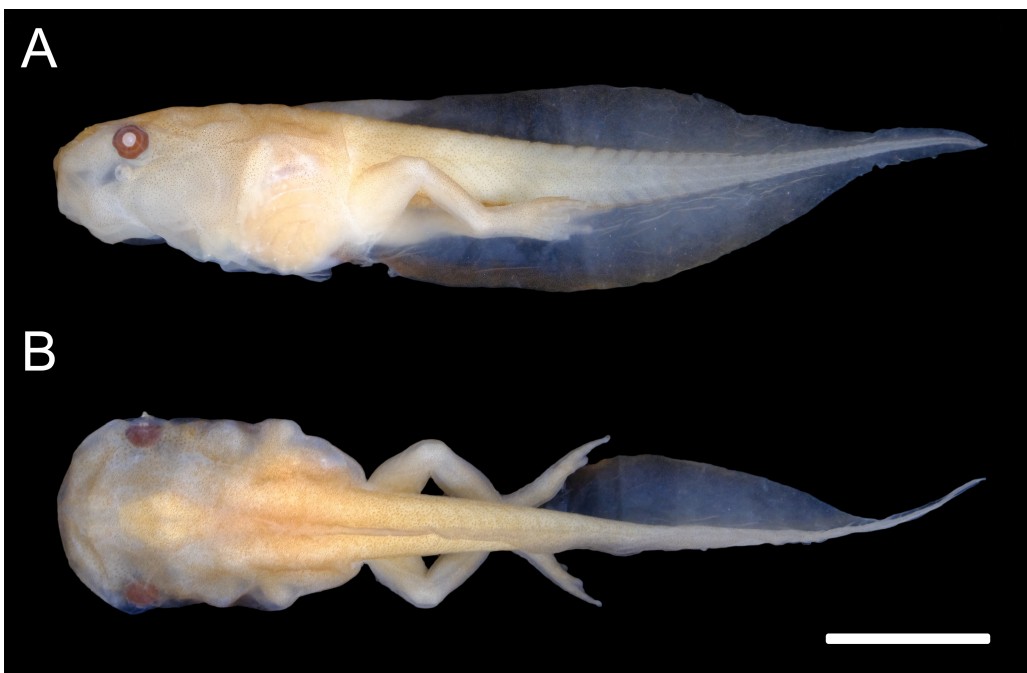

**Figure 5** External morphology of *Chiasmocleis anatipes* tadpoles at Stage 38 (KU 146836): (A) lateral; and (B) dorsal views (scale bar = five mm).

Tadpoles of *C. altomontana* had the highest values in most measurements, except for total length, eye diameter, maximum tail height and tail muscle width, which makes it one of the largest of the genus. *Chiasmocleis mantiqueira*, its sister species, had higher eye diameter and tail muscle, while *C. albopunctata* had broader tail muscle (*Oliveira-Filho & Giaretta, 2006*). The tadpole of *C. anatipes* had the largest total length (31.5 mm; *Walker & Duellman, 1974*), followed by *C. altomontana* (22.98 mm, stage 35–39), *C. mantiqueira* (19.98 mm, stage 36; *Santana et al., 2012*), *C. schubarti* (18.5 mm, stage 34–38; *Santos et al., 2015*), *C. lacrimae* (18.49 mm, stage 37; *Cordioli, Matias & Carvalho-e Silva, 2019*), and *C. leucosticta* (18.35 mm, stage 34–37; *Langone et al., 2007*). Except for *C. anatipes*, all those species belong to the Atlantic Forest clade.

The only species of the genus whose buccopharyngeal morphology was described was *C. leucosticta* (*Langone et al., 2007*), which is closely-related to *C. altomontana* within the Atlantic Forest clade. The buccal floor of the *C. leucosticta* and *C. altomontana* are quite similar, except for the number and size of the BFA papillae: *C. leucosticta* has six BFA papillae (four in *C. altomontana*). The overall characteristics are typical of other suspension feeding Gastrophryninae microhylids, such as *Elachistocleis* and to a lesser extent *Dermatonotus* (*Echeverria & Lavilla, 2000*; *Vera Candioti, 2006*; *Vera Candioti, 2007*; *Ferreira & Weber, 2021*). These include small number of papillae, median notch tall with serrated margins, a large narial valve covering the internal nares, a few prepocket papillae, and a prominent glottis with tick lips. However, both *Dermatonotus* and *Elachistocleis* have many more BFA papillae, which seem shorter and are evenly distributed along the anterior margin of the

ventral velum (*Vera Candioti, 2006*), in addition to many pustulations on the buccal floor and roof arenas. Conversely, *C. altomontana* has only four large BFA papillae and a few pustulations in that region, while lacking pustulations on the BFA and BRA. *Dermatonotus* is more closely-related to *Elachistocleis*, than to *Chiasmocleis* (*Tu, Yang & Zhang, 2018*), which might explain the similarity in their buccopharyngeal morphology.

Chiasmocleis altomontana* are explosive breeders, typical of other congeneric species (*Haddad & Prado, 2005*). The egg clutches of *C. altomontana* are deposited in bubble nests, reproductive mode 10 (*sensu Haddad & Prado, 2005*), similar to *C. leucosticta* (*Haddad & Hödl, 1997*). Other species of the genus, such as *C. lacrimae* (*Cordioli, Matias & Carvalho-e-Silva, 2019*), *C. shudikarensis* (*Menin, Souza & Rodrigues, 2011*), and *C. ventrimaculata* (*Schlüter & Salas, 1991*) lay eggs in a thick and viscous jelly clustered on the surface of the water. *Chiasmocleis hudsoni* lays their eggs on roots and tree trunks on the banks of ponds (*Rodrigues et al., 2008*), while the eggs of *C. antenori* (*Krügel & Richter, 1995*) and *C. magnova* (*Pérez-Peña, Tapia-Del-Aguila & Gagliardi-Urrutia, 2021*) develop inside terrestrial tank bromeliads. Some species that lay fewer eggs generally have larger eggs (*Peloso et al., 2014*). Egg number varies widely from six in *C. antenori* (*Krügel & Richter, 1995*) to 351 in *C. hudsoni* (*Rodrigues et al., 2008*), as well as their degree of pigmentation, from completely unpigmented in *C. magnova* (*Pérez-Peña, Tapia-Del-Aguila & Gagliardi-Urrutia, 2021*) to pigmented (*e.g.*, *C. antenori*, *Krügel & Richter, 1995*).

## CONCLUSIONS

In summary, few tadpoles of *Chiasmocleis* have been formally described (approximately one-third). Among the described species, the tadpoles of *C. altomontana* can be distinguished from its congeners by a set of external morphological traits. The morphology of *C. altomontana* is more similar to congeneric species from the Atlantic Forest, than the Amazonian clade. We provided the second description of the internal oral anatomy of *Chiasmocleis* tadpoles. However, there is still very few data available to make broader generalizations. We recommend that future descriptions should strive to include internal oral anatomy, as it provides important information for understanding ecological aspects, such as habitat and feeding, as well as for phylogenetic approaches.

## ACKNOWLEDGEMENTS

Denise de C. Rossa-Feres provided working space and essential guidance during the initial phase of this study. The ICMBio staff in the Serra da Bocaina National Park provided logistical assistance and housing. Denise D. Dias identified the intestinal contents of tadpoles. Jorge L. Nessimian (UFRJ) and his team identified the predatory insects. Katiuce Picheli kindly provided pictures of the oral apparatus of the tadpole. Ana Paula Motta kindly provided pictures of tadpoles of *C. anatipes* from Kansas University Museum. We thank Andrés Romero-Carvajal, Maria Florencia Vera Candioti, Pedro H. Dias, and an anonymous reviewer for commenting on the manuscript. Their suggestions greatly improved the final version.

### Funding

This project was funded by a FAPESP master's fellowship to Diogo Borges Provete (# 2008/55744-6) and a doctoral fellowship to Michel Varajão Garey (#08/50575-1). This study was funded by the Coordenação de Aperfeiçoamento de Pessoal de Nível Superior—Brasil (CAPES)–Finance Code 001. The work of Diogo Borges Provete has been funded by a research grant from the Brazilian National Council for Research and Technological development (Proc. # 407318/2021-6). Diogo Borges Provete received a fellowship for experienced researchers from the Humboldt Foundation during the final stages of this study. Michel Varajão Garey was funded by UNILA (editais PRPPG # 205/2021, and PRPPG # 77/2022). Tiago Leite Pezzuti received a post doc fellowship from CAPES (8887.468027/2019-00). The images were taken in a stereomicroscope acquired with a funding from Fundação de Amparo à Pesquisa do Estado de São Paulo (FAPESP) grant # 2019/09215-6. There was no additional external funding received for this study. The funders had no role in study design, data collection and analysis, decision to publish, or preparation of the manuscript.

### Grant Disclosures

The following grant information was disclosed by the authors:
FAPESP: 2008/55744-6, 08/50575-1.
Coordenação de Aperfeiçoamento de Pessoal de Nível Superior—Brasil (CAPES): 001.
Brazilian National Council for Research and Technological Development: 407318/2021-6.
Humboldt Foundation.
UNILA: PRPPG 205/2021, PRPPG 77/2022.
CAPES: (8887.468027/2019-00).
Fundação de Amparo à Pesquisa do Estado de São Paulo (FAPESP): 2019/09215-6.

### Competing Interests

Diogo B. Provete is an Academic Editor for PeerJ.

### Author Contributions

- Leandro B. C. Menezes performed the experiments, analyzed the data, authored or reviewed drafts of the article, and approved the final draft.
- Marcos R. Severgnini performed the experiments, analyzed the data, prepared figures and/or tables, authored or reviewed drafts of the article, and approved the final draft.
- Tiago L. Pezzuti analyzed the data, prepared figures and/or tables, authored or reviewed drafts of the article, and approved the final draft.
- Michel V. Garey performed the experiments, analyzed the data, authored or reviewed drafts of the article, and approved the final draft.
- Diogo B. Provete conceived and designed the experiments, performed the experiments, authored or reviewed drafts of the article, and approved the final draft.

## Field Study Permissions

The following information was supplied relating to field study approvals (i.e., approving body and any reference numbers):

ICMBio provided collecting permits (14474-1, 14861-1, and 16461-1)

## Data Availability

The data and R code are available at figshare: Menezes, Leandro; Severgnini, Marcos Rafael; Garey, Michel Varajão; Pezzuti, Tiago; Provete, Diogo (2024). The Tadpole of Chiasmocleis altomontana (Anura: Microhylidae). figshare. Dataset. https://doi.org/10.6084/m9.figshare.26084521.v3.

## Supplemental Information

Supplemental information for this article can be found online at http://dx.doi.org/10.7717/peerj.19220#supplemental-information.

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
