# Peer review of "The tadpole of Chiasmocleis altomontana (Anura: Microhylidae)"

_PeerJ, doi:10.7717/peerj.19220_

## Round 0.1 · original submission · Major Revisions

Dear authors,

Given your appeal of the former decision, the journal asked that I take over from the previous academic editor. I have decided to make a decision of 'major revisions'.

Please note, that the original reviewers may be asked to re-review your manuscript, as well as new both 'methods'-based reviewers.

I happen to agree with most of the two reviewer's comments. At present, the methodology is not clear and is not reproducible. This will need to be rectified in the revision. Principally, the code used to analyse the data is not provided, neither is the data itself (a summary of the data is provided in Table 1, but the raw data will need to be included with the revision).

I tried using the Figshare link provided, but it does not appear to be active. Can you double-check that the link does indeed work, and can you provide the data and R scripts in the next round of revision so myself and a new reviewer can double-check them?

Given that an estimated 33% of the raw data is said to be missing, and had to be inferred, I would like to see the raw data itself. I have to admit, I was surprised to read you included a dataset which had a third of its data is inferred. Especially for an ordination analysis. In most circumstances I would not accept a paper with such a high amount of missing data for this type of analysis (PCA). A full justification for using a PCA approach will need to be provided, and I would recommend the authors look into using an additional ordination method that can handle missing entries (PCord for instance). Using two different approaches for such a dataset would be more prudent.

The other issue that will need to be elucidated is the permit. Reviewer two has raised an issue regarding the date of the permit, which is over a decade after the collection of the specimens. I'm sure there is an explanation, but I am sure you can understand that PeerJ cannot publish papers without the correct permits. Given that a reviewer has highlighted this, we must ensure that the situation is clarified.

I look forward to receiving your revised manuscript.

· Appeal

Appeal


· · Academic Editor

Reject

Two expert reviewers have evaluated your manuscript and their comments can be seen below. As you will see, both point out some major issues with the experimental design, methods, data analysis and interpretation

Based on these reviews I cannot recommend that your manuscript, in its current form, be accepted due to the issues raised. However, I do encourage a resubmission of this manuscript ensuring that the reviewer´s concerns are thoroughly addressed with clear indications of where your have responded to each concern.

I am positive that the comments from both reviewers will prove very useful to you as you prepare your manuscript for further consideration by PeerJ.

·

Basic reporting

This is a very interesting article with important data from a previously unknown tadpole from the Microhylidae Family.
Overall, the text is well written, and the document is well structured. I am not an expert in this taxonomic group, nor a dedicated morphologist, however, it seems that the reviewed literature is sufficient for the introduction and discussion.
Minor suggestions:
The figures 2 and 3 need more information in the legend and the authors should label the morphological landmarks presented.
In figure 2 It is important to see differences in the early

Experimental design

Major revision required:
The authors provide insufficient, and possibly flawed, evidence to conclude that “the overall pattern suggests an evolutionary convergence in the tadpole shape, since species from distant clades were positioned at the centre of the morphospace.”
Experimental Design flaw:
My appreciation regarding this conclusion comes from the fact that there is an important amount of missing information in the matrix of linear measurements used to generate the morphospace PC analysis. Authors claim that missing info was 33% in the whole matrix, however, for some measurements, lack of data was up to 64%. The authors then use a phylogenetic tool to fill in the missing data. While it is perfectly valid to perform analyses from data extracted from the literature, where missing data is common, authors must recognize the limitations of such analyses. First, to manage data in this way, authors must declare which of the datapoints in the matrix were inferred from the phylogenetic tool. Second, authors must declare in the manuscript the limitations of inferring data in this way.

Validity of the findings

Major revision required:
The authors provide insufficient, and possibly flawed, evidence to conclude that “the overall pattern suggests an evolutionary convergence in the tadpole shape, since species from distant clades were positioned at the centre of the morphospace.”
Validity of the findings:
The authors must reassess whether it is possible to generate a morphospace PC analysis with such data quality.
In my opinion this way of generating data is not acceptable when the percentage of missing data is above 30 %. It is not acceptable because such method of imputation over large parts of the matrix (these are only 13 tadpoles) might cause artifacts in the PCA. My biggest question is how much of the centralized appearance of morphometric data is real, and how much is the result of the imputation method.
A proper assessments of the limitations in this analysis or avoiding this analysis and conclusions might help improving the quality of the manuscript.
Because of these concerns, I think this manuscript requires major revisions before being published.

Reviewer 2 ·

Basic reporting

Special attention is needed to cite the recent references.

Experimental design

Please check my comments.

Validity of the findings

Please check my comments.

Additional comments

Dear authors,
Given the scarcity of information on tadpoles, this study is very important. Describing the morphology of anuran larvae fills essential gaps in our understanding of taxonomy and natural history, especially for species with elusive adult stages. The focus on Chiasmocleis altomontana, an explosive breeder, provides valuable insights.
In general, the manuscript is well-written and well-structured. However, there are several major and minor flaws that need to be addressed to improve the clarity, robustness, and scientific rigor of the study.
Major Flaws:
1. The identification of the species did not follow any standard method. Proper identification protocols, including genetic confirmation or detailed morphological keys or bioacoustics, are crucial for ensuring the accuracy of the findings.
2. Some parts of the methods section are vague and unclear. Specifically, the authors did not mention whether they used photos of live specimens or preserved specimens for the morphological measurements. It is also not clear which specimens were used for the anatomical analysis. Additionally, the date of permit, apparently in 2021, and the period of study, 2008 – 2010, were elusive.
3. The discussion section includes only comparisons with congeners and lacks robust analysis and reasoning of the similarities and/or dissimilarities of the current results. A more thorough discussion, integrating broader ecological and evolutionary contexts, would strengthen the manuscript.
Minor Flaws:
1. The title of the manuscript does not accurately reflect the content and focus of the study.
2. Some parts of the manuscript present information in a non-sequential manner, which can confuse readers. A more logical flow of information is recommended.
3. Authors misused the subjective terms, especially the terms larva and tadpoles.
4. The manuscript inconsistently uses full names and abbreviations for genera. Consistency in this area is necessary.
However, other minor corrections and comments have been provided directly in the manuscript.

---

## Round 0.2 · Minor Revisions

Dear authors,

Thank you for your revised manuscript. Unfortunately, I was unable to get the same reviewers as round 1 to re-review your manuscript. I also was unable to get more than one reviewer this round. Based on their comments on your text and response to reviewers.doc, as well as my own examination, I am making a 'minor revisions' decision.

I believe that you contacted Pete Binfield about the permit situation? Can you include a shortened version of what you told Pete in your manuscript? That will rectify the issue, and allow us to proceed to acceptance.

I would also like to thank you for positively engaging with the reviewers comments and addressing them all.

Finally, in regards to your comments that the phylomorphospace being a 'dataviz' technique - ordination analyses are used to analyse datasets. The accompanying ordination plot, or 'morphospace', being a euclidean visual representation of the higher dimensional axes. Given that you have taken on board the comments I am ok with the analysis being presented.

I look forward to receiving your revised manuscript.

Reviewer 3 ·

Basic reporting

Materail and methods are well explained and the description of the tadpole is very good.

Experimental design

no comment

Validity of the findings

The paper is a nice contribution to tadpole biology and natural history.

Additional comments

This is a well-written manuscript. The description of the tadpole of Chiasmocleis altomontana is well-done and detailed and it will contribute to understanding the variation of tadpole morphology in the genus.
I carefully read the authors reply to the reviewers and the I found it detailed and clear.
Overall, I do not find minor or major problems with the manuscript.
In my assessment, this a very nice contribution to the diversity of tadpoles.

---

## Round 0.3 · accepted · Accept

Dear authors,

Firstly, let me apologise for the upset over the handling of this manuscript. I think it would have benefitted from better initial review, but I note that the initial editor struggled to find any or appropriate reviews. As I'm sure you are aware, this is a growing problem globally and not just with PeerJ.

Both of the most recent reviews refer back to their original reviews (in round 1) without adding any further information. Therefore, I considered only the comments given in round 1 of review referring to the latest version of this manuscript.

My own reading of this manuscript is that it is an excellent description and morphological analysis of the tadpoles of Chiasmocleis altomontana.

I apologise again for any upset that the authors have experienced during the handling of this manuscript. This is the reason why PeerJ has section editor oversight.


For the record, I note below how the authors have adequately handled the comments of both reviewers in the initial round of review. Line numbers are left from the most recent version of the manuscript:

Major Flaws:
1. The identification of the species did not follow any standard method. Proper identification protocols, including genetic confirmation or detailed morphological keys or bioacoustics, are crucial for ensuring the accuracy of the findings.

>>Unfortunately, we did not
111 collect any tissue samples to conduct molecular identification of the tadpoles at the time of
112 collection. Given that the lot of tadpoles has been stored in 10% formalin for more than 10
113 years, it was not possible to extract relevant DNA material for a molecular identification either.
114 However, it is noteworthy that Chiasmocleis altomontana is the only species of this genus
115 known to occur in the area in more than 60 years of surveys (Zaher et al., 2005; Serafim et al.,
116 2008; Garey et al., 2014).

2. Some parts of the methods section are vague and unclear. Specifically, the authors did not mention whether they used photos of live specimens or preserved specimens for the morphological measurements. It is also not clear which specimens were used for the anatomical analysis. Additionally, the date of permit, apparently in 2021, and the period of study, 2008 – 2010, were elusive.

>>Both adults and tadpoles we found in this
122 pond were collected and are now housed in the DZSJRP and CFBH collections (Appendix 1).
123 ICMBio provided collecting permits (#14474-1, 14861-1, and 16461-1). At the time of sampling
124 (2008 and 2009), the Brazilian law did not require any further authorization as institutional
125 animal care and use committees were not established yet, given that the National Council for
126 Animal Experimentation ñ CONCEA was created by law in October 2008 (Law # 11.794, of 8
127 October 2008). Thus, field work, sampling, and transportation followed the IBAMA ordinance
128 (Portaria #16 of 4 March 1994).
Description of the larval external morphology was based on 11 specimens between
130 stages 35 ñ 39 (Gosner, 1960).

3. The discussion section includes only comparisons with congeners and lacks robust analysis and reasoning of the similarities and/or dissimilarities of the current results. A more thorough discussion, integrating broader ecological and evolutionary contexts, would strengthen the manuscript.
>>This is entirely appropriate for this paper.


Minor Flaws:
1. The title of the manuscript does not accurately reflect the content and focus of the study.
>>Null comment

2. Some parts of the manuscript present information in a non-sequential manner, which can confuse readers. A more logical flow of information is recommended.
>>Null comment

3. Authors misused the subjective terms, especially the terms larva and tadpoles.
>>Null comment

4. The manuscript inconsistently uses full names and abbreviations for genera. Consistency in this area is necessary.
However, other minor corrections and comments have been provided directly in the manuscript.
>>Null comment

Major revision required:
The authors provide insufficient, and possibly flawed, evidence to conclude that “the overall pattern suggests an evolutionary convergence in the tadpole shape, since species from distant clades were positioned at the centre of the morphospace.”

>>Neither the term "evolutionary convergence" nor "convergence" occurs in the most recent version of this manuscript.

Reviewer 2 ·

Basic reporting

.

Experimental design

.

Validity of the findings

.

Additional comments

I am still not satisfied with the responses from the authors (the attached rebuttal letter). Most of the responses were inappropriate and did not address the issues raised. Furthermore, they did not reply to the comments embedded directly in the manuscript, which had many suggestions and corrections. The language and consistency in using the technical terms (scientific names and others) did not meet the standard of a quality manuscript. If I were the editor, I would prefer to ask the authors to resolve the issues and follow the comments made by the reviewers first to ensure the quality of the manuscript before accepting it for publication.

Annotated reviews are not available for download in order to protect the identity of reviewers who chose to remain anonymous.

Reviewer 3 ·

Basic reporting

The manuscript is clear and provides novel information.

Experimental design

N/A

Validity of the findings

Clear and informative description of the larvae of Chiasmocleis altomontana